# PGT-SR: A Comprehensive Overview and a Requiem for the Interchromosomal Effect

**Darren K. Griffin** [1,*] and **Cagri Ogur** [2,3]

1    School of Biosciences, University of Kent, Giles Lane, Canterbury CT2 7NJ, UK
2    Department of Bioengineering, Yildiz Technical University, 34349 Istanbul, Türkiye
3    Avrupa Laboratories, Igenomix Turkey, 34384 Istanbul, Türkiye
*    Correspondence: d.k.griffin@kent.ac.uk

**Abstract:** Preimplantation genetic testing for structural rearrangements (PGT-SR) was one of the first applications of PGT, with initial cases being worked up in the Delhanty lab. It is the least well-known of the various forms of PGT but nonetheless provides effective treatment for many carrier couples. Structural chromosomal rearrangements (SRs) lead to infertility, repeated implantation failure, pregnancy loss, and congenitally affected children, despite the balanced parent carrier having no obvious phenotype. A high risk of generating chromosomally unbalanced gametes and embryos is the rationale for PGT-SR, aiming to select for those that are chromosomally normal, or at least balanced like the carrier parent. PGT-SR largely uses the same technology as PGT-A, i.e., initially FISH, superseded by array CGH, SNP arrays, Karyomapping, and, most recently, next-generation sequencing (NGS). Trophectoderm biopsy is now the most widely used sampling approach of all PGT variants, though there are prospects for non-invasive methods. In PGT-SR, the most significant limiting factor is the availability of normal or balanced embryo(s) for transfer. Factors directly affecting this are rearrangement type, chromosomes involved, and sex of the carrier parent. De novo aneuploidy, especially for older mothers, is a common limiting factor. PGT-SR studies provide a wealth of information, much of which can be useful to genetic counselors and the patients they treat. It is applicable in the fundamental study of basic chromosomal biology, in particular the purported existence of an interchromosomal effect (ICE). An ICE means essentially that the existence of one chromosomal defect (e.g., brought about by malsegregation of translocation chromosomes) can perpetuate the existence of others (e.g., de novo aneuploidy). Recent large cohort studies of PGT-SR patients seem, however, to have laid this notion to rest, at least for human embryonic development. Unless new evidence comes to light, this comprehensive review should serve as a requiem.

**Keywords:** PGT-SR; chromosome; translocation; inversion; Robertsonian; reciprocal

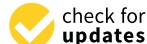



## 1. Introduction

Structural chromosomal rearrangements (SRs) are either observed as segmental aneuploidy (extra or missing parts of chromosomes) when unbalanced or, when balanced, usually presenting as a relatively normal phenotype, only discovered when the individual tries to conceive [1]. In other words, carriers of SRs are prone to infertility, repeated miscarriage, and recurrent stillbirth as well as babies with severe congenital disorders including developmental delay [2]. Men with balanced SRs may present with compromised semen parameters and the subsequent fertility issues that can ensue [3]. The fertility problems that can arise when either partner has an SR are variable (indeed they can even vary when siblings carry the same SR). Factors that can also contribute to the infertility phenotype include rearrangement type (translocation, inversion, etc.), the chromosome(s) in question, and chromosomal breakage points. Other factors that have been discussed include sex, age, family history, semen parameters, ovarian reserve, and the presence of an interchromosomal effect (ICE) [4]. ICE is a concept based on an observation that goes

as far back as 1963; suggesting that structural abnormalities may affect the segregation of other chromosomes that are not involved in the rearrangement, increasing the likelihood of whole chromosomal or segmental abnormalities [5]. Prenatal diagnosis is, of course, available to carriers; however, like all forms of PGT, PGT-SR exists for couples who do not feel they can have an elective termination of a fetus affected with segmental aneuploidy. The first "patient work-up" of a putative PGT-SR patient appears in a Delhanty-supervised Ph.D. thesis [6] and the original fluorescence in situ hybridization (FISH) strategy (see later) was, we believe, a Delhanty invention. PGT-SR by FISH was, for some time, a relatively mainstream activity and, unlike other forms of PGT, a relatively uncontroversial variant of IVF treatment. The first multi-center analysis collated by the European Society of Human Reproduction and Embryology (ESHRE) consortium included 4253 treatment cycles for the first ten years of PGT-SR (1997 until 2007) [7]. Since 2007, collating consortium numbers has become impractical but, as new technologies such as array comparative genomic hybridization (aCGH) and next-generation sequencing (NGS) come online, numbers are undoubtedly increasing. The various biopsy techniques are common to all forms of PGT and the diagnostic techniques (in addition to FISH, aCGH, and NGS) also included single-nucleotide polymorphism (SNP) arrays (chips). For detailed overviews, see the following: [8,9].

## 2. Balanced SRs in Overview

SRs arise as a result of chromosomal breaks that subsequently rejoin in a different chromosomal location [10]. Derivative chromosomes thereby ensue, rearranging the gene order and linkage patterns. Any chromosome may participate in an SR; however, certain genomic regions are more prone to breakage. Mechanisms include DNA double-strand break repair employing non-homologous end-joining (NHEJ) [10], non-allelic homologous recombination (NAHR) [11,12], chromothripsis [1,13], microhomology-mediated break-induced replication (MMBIR), and fork stalling and template switching (FoSTeS) [14]. Also implicated is chemical or radiation exposure, and the nuclear organization of the chromosome territories [15]. The types of chromosome rearrangement usually found for PGT-SR cases are covered in the next section and individuals usually inherit SRs from their parents [16,17]. The rate of inherited balanced SR is thought to be around 0.27% in fetuses (assayed by prenatal diagnosis) compared to de novo balanced SRs at 0.076–0.096% [16].

## 3. Robertsonian Translocations

Robertsonian translocations (RobT) are formed by the fusion of two acrocentric chromosomes (13, 14, 15, 21, and/or 22) reducing the number of chromosomes in the karyotype to 45. The reported RobT incidence is 1/1085 live births [2,3]. During the first meiotic division, derivative and normal chromosomes form a trivalent configuration leading to either alternate, adjacent, or, rarely, 3:0 segregation patterns. This is depicted in Figure 1.

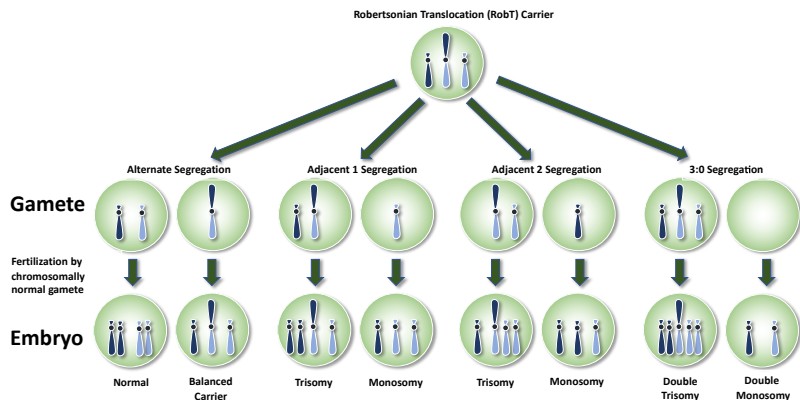

**Figure 1.** Segregation patterns of a Robertsonian translocation (RobT) in the gametes and in the resulting embryos following fertilization with a normal gamete.

## 4. Reciprocal Translocations

Reciprocal translocations (RecT) arise when non-homologous chromosomes exchange and produce two derivative (der) chromosomes. Their incidence is estimated to be 1 in 600 (0.17%) liveborns [18] but much higher (5.7%) among couples that suffer recurrent miscarriages [2]. During meiosis, the two derivatives, plus the two homologous unaffected chromosomes, adopt a quadrivalent conformation. Segregation patterns during meiosis I are similar in principle to RobTs except that five major segregation outcomes are possible: alternate, adjacent-1, adjacent-2, 3:1, and 4:0, leading to at least sixteen outcomes (Figure 2). If we factor in recombination events within the quadrivalent, leading to asymmetric segregations at meiosis II, around 32 different gametes could be produced. Notably, two that arise from alternate segregation give rise to chromosomally balanced embryos [19].

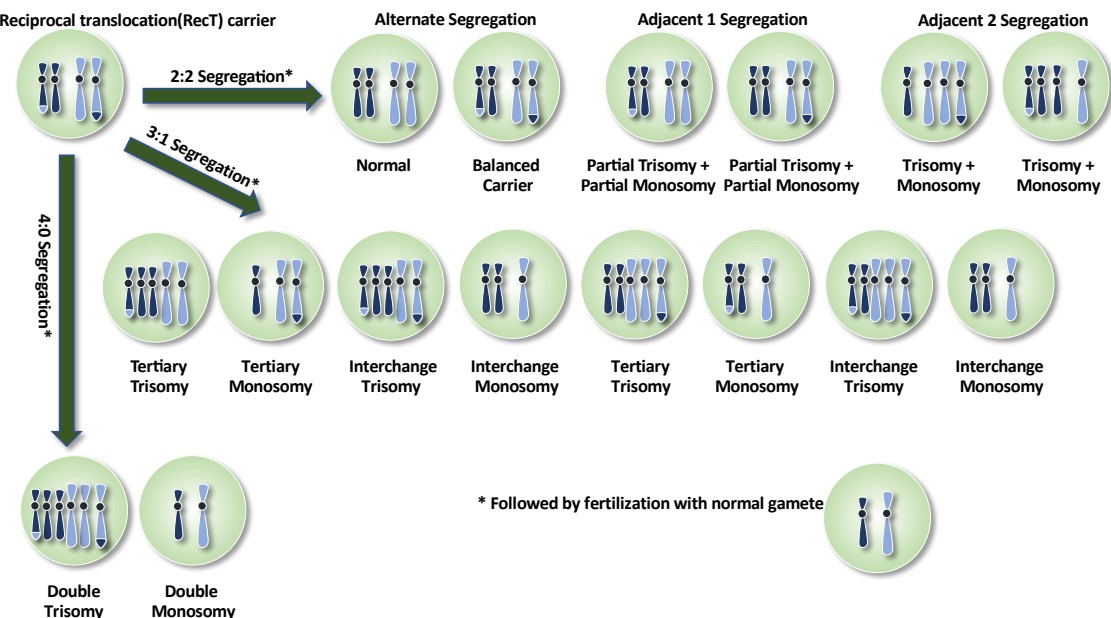

**Figure 2.** Segregation patterns of a reciprocal translocation (RecT) i in the resulting embryos followed by fertilization with a normal gamete.

Oocytes are hard to come by for research purposes and thus rarely studied; therefore, the majority of works on chromosome segregation of RecTs and RobTs use sperm. Human–hamster sperm–oocyte fusion to achieve karyotyping [20] was superseded by fluorescence in situ hybridization (FISH) because of the relative technical simplicity and ability to look at a much greater number of cells [21]. Comparative limitations, however, include the number of chromosomes that can be analyzed and difficulties in evaluating the signals. Comprehensive chromosomal analysis of sperm is also possible using techniques such as aCGH and NGS, however, there are significant cost implications [22]. In one such study, 43 individual spermatozoa from a man carrying a balanced reciprocal translocation between chromosomes 2 and 12 have been analyzed using the aCGH technique. Segmental aneuploidy pertaining to the translocation regions was observed in 18.6% of the gametes, compared to alterations in other chromosomes in 16.3% [22]. A remarkable heterogeneity in the proportion of unbalanced sperm and segregational characteristics was observed between studies. For instance, analyzing 136 reciprocal translocation heterozygotes in various studies led to the detection of unbalanced spermatozoa at a range of 19–91% of gametes [23]. A further factor is that reduction of genetic recombination within the pairing cross decreases the proportion of balanced/normal gametes [24]. The same quadrivalent can display different chiasmata distributions varying from person to person. In other words, each RobT/RecT carrier carries an individual risk in terms of their reproductive outcome. Therefore, for PGT-SR, patients should be properly advised by a genetic counselor, with ref-

erence to the outcomes that could arise as a result of their SR. The percentage of unbalanced gametes in RobT/RecT carriers could then be used to assist in the calculations of predictive value on embryonic outcomes in which there appears to be a strong correlation [25].

## 5. Inversions

Inversions involve two breaks within the same chromosome and subsequent re-orientation of the resultant piece in the opposite direction (with a change in gene order). Pericentric inversions (PEI) span the centromere, whereas paracentric inversions (PAI—incidence 0.1–0.5%) do not [26]. As with translocations, considerable reproductive risks are associated with PAI and PEI. Severity depends on the chromosomes involved and the size and distribution of the inverted segment. The meiotic configuration associated is an inversion loop. In PAIs, an odd number of chiasmata in the loop leads to acentric fragments (which are lost eventually) and dicentric bridges. In PEIs, partial duplication and deletion of the distal parts of the inversion lead to segmental aneuploidy [27]. PAIs produce fewer unbalanced gametes; in general terms, recombination is reduced in the inversion loop [28,29] and the proportion of unbalanced gametes may be associated with segment size [30,31]. The proportion of unbalanced gametes arising from inversion carriers can be as little as zero, up to slightly over 50% [23].

The most commonly reported inversion is the chromosome 9 PEI, specifically an inversion of the heterochromatic region from the q to the p arm of the chromosome, believed to occur in between 0.25% and 3.5% of the population (presumably dependent on the population studied [32]). Whether this is an inconsequential variant or clinically significant is up for debate, with some studies conflicting. Associations with infertility, repeated pregnancy loss, cancer, congenital abnormalities, and growth retardation, however, appear in the literature [33].

In general terms, when considering inversions as a whole, the risk to reproductive outcome is dependent on the chromosome involved, the size of the inverted segment, and the recombination rate within the pairing loop [30,31,33]—all factors for a genetic counselor to take into account.

## 6. Insertional Translocations and Complex Chromosomal Rearrangements

Balanced insertional translocations arise via the introduction of part of a chromosome into another non-homologous counterpart. As with other types of SR, sperm FISH might provide a useful guide to assess the risk for male carriers, assisting genetic counselors [34]. The rarity of this type of SR, however, means that meaningful data is limited [35,36].

Complex chromosomal rearrangements constitute SRs not involved in any of the above. They can be double translocations, triple, quadruple, etc. [37]. In the case of triple SRs, hexavalent meiotic configurations form [37]—the chances of balanced gamete formation reduce with increasing complexity, presenting significant challenges to genetic counselors advising PGT-SR patients.

## 7. Practicing PGT-SR

PGT-SR is offered to SR-carrying patients as an alternative to prenatal diagnosis. Numerous sampling approaches and genomic analysis techniques have been used to isolate normal (or at least chromosomally balanced) embryos. PGT-SR techniques mostly benefitted from the advancement of PGT-A techniques, which use similar technologies. The following sections summarize these.

## 8. Invasive Sampling Methods for PGT-SR

Sampling methods include polar body biopsy, blastomere (cleavage-stage) biopsy, and trophectoderm (blastocyst-stage) biopsy [7,8,38], each with its unique pros and cons. Polar body biopsy can only be used for maternal carriers and is especially useful when legislation in certain countries precludes embryo analysis [38,39]. Cleavage-stage biopsy became the most popular method until 2005, but has since been discontinued due to disadvantages such as a limited number of cells available for analysis, increased risk of misdiagnosis due to high rates of mosaicism, and possible damage to the embryo. For nearly more than one decade, trophectoderm biopsy is predominantly used not only for its many advantages in terms of cost, and availability of a greater number of cells for analysis, but also for being less likely to damage the embryo [40]. However, one recent study claims that when performed correctly, the impact of biopsy and subsequent vitrification of the cleavage-stage embryos are similar to blastocysts in terms of cell viability, spindle/chromosome configuration, and ultrastructural safety of the organelles of the embryo [41].

## 9. Minimally Invasive Sampling Methods for PGT-SR

Blastocoel fluid or spent medium act as alternative sources for DNA sampling [42,43]. Blastocentesis concordance rates with trophectoderm samples are thought to be around 97%; however, amplification rates (~82%) are far from optimal [42]. Assessment of spent culture medium for PGT-SR has been reported as successful in a male RobT (14:15) carrier resulting in a live birth of a karyotypically normal, healthy child [44]. Jiao et al. [45] performed PGT-SR on mixed blastocoel fluid/spent medium, trophectoderm, and whole embryo samples using multiple annealing and looping-based amplification cycle (MALBAC) A total of 41 blastocysts from 22 couples with SRs were successfully analyzed for segmental aneuploidy, achieving a high resolution (~1 Mb) without maternal contamination [45].

## 10. FISH for PGT-SR

FISH is the original diagnostic approach used for PGT-SR, and is less controversial than PGT-A, as it is a targeted approach. To the best of our knowledge, it was Joy Delhanty who first devised the classic detection protocol for FISH-based PGT-SR, involving the identification of usually centromeric and sub-telomeric regions with fluorescent probes. This, in turn, was an adaptation of the pioneering work using X- and Y-chromosome probes for sex selection [46], and for aneuploidy [47]. The strategy mostly involved cytogenetic analysis of interphase cells, with a three-color strategy (e.g., with one centromeric and two sub-telomeric or two centromeric and two sub-telomeric probes) [48–50].

Such a protocol is, however, limited by the need for pre-clinical work-up on metaphase chromosomes of peripheral blood carriers to confirm the breakpoints and test the efficiency of probes, which is time-consuming and expensive. The need for fixation of blastomeres or trophectoderm cells also limits the procedure, as a good quality nucleus free of surrounding cytoplasm depends on the technical skills of the operator to avoid non-specific signals, suboptimal hybridization, and auto-fluorescing artifacts. Several fixation methods are published, but the most common [51] involves hypotonic solution and 3:1 methanol-acetic acid. A major limitation of this approach is the inability to detect chromosomes not involved in the rearrangement [52] and thus FISH for PGT-SR is mostly now obsolete. It nonetheless still has a place for cryptic translocations and inversions when breakpoints are <2 Mb from the telomere.

Until comprehensive chromosomal analysis techniques, such as array-based and NGS systems came along, researchers tried to improve the informativity of the FISH system both quantitatively (adding extra probes for aneuploidy) and qualitatively, such that it could also distinguish a balanced chromosome complement from a normal one. For this reason, carrier-specific probes were developed to be used in interphase cells [48]. This approach was based on the hybridization of breakpoints usually spanning commercial DNA probes (see Figure 3); however, it requires a major pre-clinical work-up, developing those probes for each specific translocation. Another approach was using FISH on metaphase chromosomes from single blastomeres using conversion via nuclear transfer or chemical solutions [53,54]. In order to create metaphase preparations, single blastomeres were fused with enucleated or intact mouse zygotes (nuclear transfer) or were treated with caffeine and colcemid [54]. The nuclear transfer technique was applied to 437 blastomeres, of which 88% resulted in successful nuclear conversion and 29% in clinical pregnancy rate per transfer, with 7 healthy deliveries in 52 cycles [53]. In a larger study, including the results of the previous experience, a chemical conversion method was applied to 946 blastomeres in 94 cycles with 71% efficiency, leading to a conversion rate that decreased spontaneous abortion by 13–15% compared to their PB1/PB2 control group (25%) [54]. Nevertheless, this technique is labor-intensive, limited by the availability of fertilized mouse zygotes and the efficiency of the conversion method.

Most of the PGT-SR with FISH was performed without additional aneuploidy screening [55,56] and thus only a few have yielded results of chromosomal abnormalities unrelated to the parental error [57–59]. The rate of abnormalities not involved in translocations has been found to be relatively frequent in embryos obtained from RobT carriers and was higher than in RecT carriers (67% vs. 22%) [57]. In another study, an overall 60.3% aneuploidy rate was found after the analysis of five chromosomes (13, 16, 18, 21, and 22) in cleavage-stage embryos of 13 RecT carriers [58]. Only 8.7% of them were both normal/balanced or euploid for the five chromosomes, and thus transferable. Here, the aneuploidy rate was similar when compared between genders and between age groups (≤37 and >37 years old) but differed among normal/balanced embryos and unbalanced embryos. Aneuploid embryos were more likely to have an unbalanced complement, suggesting a global disruption of mitotic and meiotic segregations of chromosomes. Embryo transfer was performed for nine carriers but no clinical pregnancy was achieved in this cohort. In another study, aneuploidy screening was performed in RecT carriers for chromosomes 13, 18, 21, X, and Y, and revealed that 46.8% of 141 embryos were aneuploid. The live birth rate was 26.7% per embryo transfer; however, the authors concluded that additional aneuploidy screening did not improve the clinical outcomes [59].

The reproductive history of rearrangement carrier couples before PGT-SR treatment is mostly unsuccessful with live-born delivery rates of around 4.8–9.7% [59]. Numerous manuscripts claim to observe a live birth rate increase and a miscarriage rate decrease after PGT-SR when compared to normal (non-assisted) conception. Munne et al. [49] reported PGT-SR in 35 cases, demonstrably improving spontaneous abortions from 92% in natural conceptions to 12.5% for PGT-SR cycles ($p < 0.001$). This was mirrored in further studies post-PGT-SR, a 24.7% implantation rate, plus an 18.6% miscarriage rate was observed, improving the "take-home baby rate" from 11.5% to 81.4% [53] and to 85.7% [49].

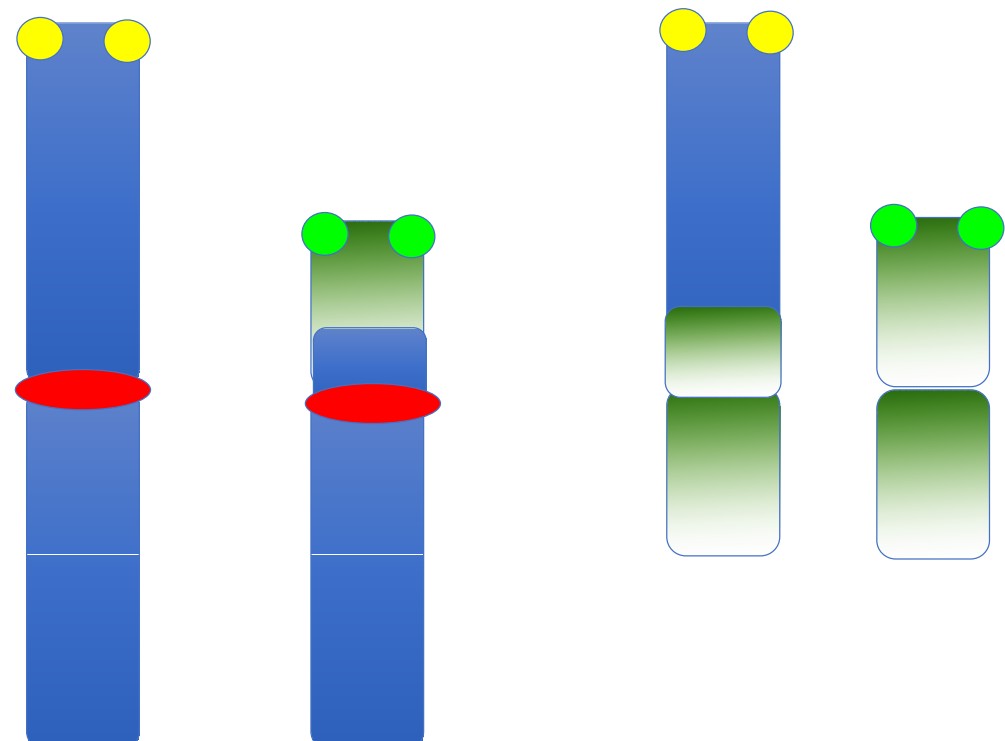

**Figure 3.** FISH detection protocol probably first devised by Joy Delhanty. At least three probes are required, two of which must be on either side of one of the chromosomal breakpoints. The availability of commercial probes in the late 1990s and early 2000s meant that centromeric and sub-telomeric probes were often used, typically in red, green, and yellow colors. Balanced and unbalanced segregation patterns could be observed in the interphase nuclei of fixed preimplantation methods. Although popular and effective, this approach was eventually superseded by comprehensive chromosome screening methods such as aCGH, SNP arrays, and NGS.

Despite these rare, successful studies, attempts to improve FISH were limited, possibly due to the prevailing use of cleavage-stage biopsy at the time [40]. The ESHRE PGD consortium data reported an overall clinical pregnancy rate of 12–17% per oocyte retrieved and 22–26% per embryo transferred [6]. Using a systematic review approach, reproductive success with PGT-SR was compared with natural conceptions in similar balanced chromosome rearrangement carriers [60]. This review encompassed four studies with 469 cases of natural conception plus 21 studies with 126 PGT-SR cases. The cumulative live birth rate was 33–60% (median 34%) in the natural conception group, whereas after PGT-SR, it was a maximum range of 0% to 100% (median 31%). The miscarriage rate showed a modest, but not statistically significant, improvement, namely 21–40% (median 34%) in natural conception and 0–50% (median 0%) following PGT-SR. A further meta-analysis [61] showed similar live birth rates, time to first conception, and miscarriage rates seen in natural conception compared to PGT-SR in couples with recurrent pregnancy loss (studies from 1997 to 2014). It is noteworthy that these studies were limited to couples who had had at least two or more miscarriages.

## 11. STR-Typing for PGT-SR

Short tandem repeat (STR) typing comprises PCR-based methods and multiplex STR markers located on either side of the chromosomal breakpoint [62]. Thorough work-up is required, for each of the cases, unique to the chromosomal rearrangement of interest. Traversa et al. [62] analyzed 29 couples using this method, finding that the proportion of alternate segregation for RecT was 33% and 77% for RobT. Fetal heartbeat rate was 40% (RobT carriers) and 46% (RecT carriers). The approach benefits from providing controls for exogenous DNA contamination and detection of uniparental disomy [63]. This study looked at 241 embryos from 27 couples finding the proportion of alternate segregations was 38.5% for RecT and 66.1% for RobT carriers. Ninety embryos were also analyzed for copy-number changes in chromosomes 13, 14, 15, 16, 18, 21, 22, X, and Y, finding a 63.1% aneuploidy rate. An implantation rate of 59.6% was also established [63].

## 12. Comparative Genomic Hybridization (CGH) and Array Comparative Genomic Hybridization (aCGH)

Both FISH and STR-typing were ultimately replaced by array-based methods such as aCGH and SNP arrays. It thereby became feasible, for the first time, to detect aneuploidy and segmental imbalances of all the chromosomes [64,65]. Comparative genomic hybridization (CGH) was primarily introduced as a means of detecting somatic chromosome loss and gain in cancer cells [66]. When applied to embryo biopsies, DNA from single (or small numbers of) cells were amplified using whole-genome amplification (WGA), co-hybridization of red and green fluorescently labeled test and reference DNA preceded application to chromosome preparations and epifluorescence microscopy analysis [67]. The low resolution of this approach (10–25 Mb) [64,68], as well as its laborious and time-consuming nature, meant that its use was limited. Nonetheless, Malmgren et al. [64] looked at 94 biopsied blastomeres from seven couples who were carriers of SRs using this approach. The confirmation rate between CGH and FISH was low, possibly due to near-universal mosaicism and a number of cells with a chaotic chromosome complement. Chromosomal CGH techniques were soon superseded by microarray-based CGH (aCGH), an altogether more simple and automated method [69]. Metaphase chromosome preparations were thus replaced by small dots of DNA (bacterial artificial chromosomes (BACs) or with oligonucleotide sequences. Bespoke software analysis programs assess the red:green ratios, thereby detecting chromosomal losses and gains. An example of how this was analyzed is the BlueFuse Multi software (Illumina, San Diego, CA, USA), which determines the median log2 ratio for each chromosome (and each chromosomal segment). The resolution is 5–10 Mb for BAC arrays (or occasionally 2.5 Mb [70]) to 20–50 kb using oligonucleotide arrays [71], but balanced rearrangements cannot be identified. In a paper comparing chromosomal CGH, aCGH (BACs), and aCGH (oligonucleotides) analyzing cleavage-stage embryos, all three approaches gave similar profiles, albeit with the oligo-array providing the highest resolution (~20 kb) [71].

aCGH also permits simultaneous detection of other de novo losses and gains, unrelated to the chromosome abnormality of the carrier parent [52,70,72]. It is thus an altogether more effective method than interphase FISH, given that around a quarter of embryos that are a result of balanced segregations have an additional chromosome abnormality.

FISH as the "gold standard" for PGT-SR, therefore, was ultimately discontinued in favor of more comprehensive chromosomal screening approaches. One of the earliest studies [70] using FISH achieved a 70.6% clinical pregnancy rate plus a 63.6% implantation rate in a cohort of translocation carrier patients. The first healthy live birth was reported later that year [73] involving 20 cycles (five RobT and nine RecT and two inversion carriers) with a resolution of detection below 3 Mb. All biopsy methods were included (polar body, blastomere, trophectoderm) with 91.8% WGA efficiency. Only 22.3% of embryos were because of chromosome abnormalities involving both the chromosomes of interest (27.3%) and de novo imbalance (28.9%) and 27.3% with both. The cumulative pregnancy rate per embryo transfer was 45.5% and the live birth rate per embryo transfer was 27% with no miscarriages [73]. Christodoulou et al. [72] studied 34 PGT-SR couples (50 cycles, 9 RobTs, 21 RecTs, 2 inversions, 1 insertional translocation, and 1 complex translocation) using trophectoderm biopsy as a starting material. A total of 35.7% of embryos were normal or balanced overall. Regarding the 133 abnormal embryos, 36.1% had an abnormality that arose as a result of malsegregation of the chromosomes rearranged in the carrier parent. Ghevaria et al. [52] established that 55–65% of cleavage-stage embryos (22 cycles, 16 RecT carriers, 7 RobT carriers) displayed extra aneuploidies of chromosomes not involved in the translocation. Subsequent FISH follow-up demonstrated that meiotic aneuploidy was present in 35% of embryos, 47% had mitotic errors, and 15% had both; 63% carried additional de novo chromosomal imbalance. Fodina et al [74] analyzed chromosomal differences in terms of translocation type and of carrier sex in of 10 couples finding the lowest aneuploidy rate in the male carrier group and the highest in the Robertsonian translocation carrier group. The prevalence of chromosomal aberrations was 4.5× greater in the reciprocal, compared to the Robertsonian translocation carrier group. with 4.7× higher aneuploidy rates in female compared to male carrier groups.

PGT-SR, therefore, benefitted from the success of the aCGH era for PGT-A, improving the cumulative pregnancy rate from 40% with FISH [6] to 62%. aCGH, however, eventually gave way to SNP arrays and NGS

## 13. Karyomapping and SNP Arrays for PGT-SR

A single-nucleotide polymorphism (SNP) is defined as a DNA sequence variant that occurs every one in 1000 nucleotides. A PGT-SR diagnostic SNP microarray would normally consist of around 300,000 features [74]. Assaying parental DNA precedes the establishment of four parental haplotypes for each chromosome region and the subsequent detection of parental origin. Polymorphic genotypes are denoted as AA, AB, and BB at each locus and analyzed in comparison to the human HapMap reference. SNP arrays tend to be denser than CGH arrays, thereby providing higher resolution. SNP arrays were also allowed to distinguish between balanced carriers and normal embryos [75,76]. One possible drawback of this approach is the availability of parent DNA and at least one unbalanced embryo as a reference [77]. To distinguish carrier and normal embryos, informative SNPs within 5 Mb of the chromosomal breakpoints of each chromosome involved in the rearrangement are required [77]. Comparing genotypes at informative SNP loci can also detect uniparental disomy. Karyomapping, base phasing, and haplarhythmisis are advanced techniques based on SNP detection throughout the genome [75,78,79].

Treff et al. [80] looked at 18 couples with balanced RobTs and RecTs treated by PGT-SR following trophectoderm biopsy and SNP array analysis. An implantation rate of 45% along with a cumulative pregnancy rate per embryo transfer of 75% was achieved. This work highlighted the value of simultaneous screening for chromosomes not involved in the chromosome rearrangement of the carrier parent. From 122 embryos analyzed, 62 were normal/balanced, with the remaining 23 being aneuploid for another chromosome. van Uum et al. [81] looked at 36 cleavage-stage embryos "cell by cell" that had previously been determined unbalanced by FISH. Subsequent SNP array analysis revealed concordance with the primary FISH diagnosis: 64% were confirmed, 14% were balanced (opposite to the initial diagnosis), and 22% displayed mosaicism. Tan et al. [82] compared the efficacy of

SNP arrays performed on blastocyst embryos compared to FISH on cleavage-stage embryos. The approach that employed SNP array demonstrated greater implantation rates (69% compared to 38% for RobT and 74% compared to 39% for RecT) ($p < 0.001$). Indeed, the SNP-based approach identified more than 15% more chromosomal abnormalities. Moreover, the percentage of transferable embryos was greater using SNP-based approaches. Overall, SNP array approaches are thought to outperform the FISH methods in terms of higher pregnancy rates (cumulative pregnancy rates of 45–70% per transfer [70,78,80]). In a comparison of SNP arrays with aCGH platforms, Tobler et al. [83] looked at day three and day five embryos from RecT carriers, observing statistically significant differences in the proportion of chromosomally balanced embryos, but not in the overall clinical pregnancy rates (60% for SNP arrays, 65% for aCGH). From nearly 500 embryos analyzed, 45% were chromosomally balanced; 24% were balanced or normal for the chromosomes rearranged in the carrier parent, but aneuploid for other chromosomes; 23% only had an imbalance of the chromosomes involved in the parent's rearrangement, with 8% containing imbalance for both. It was reported that SNP arrays outperformed aCGH, detecting 47% euploid/balanced embryos compared to 39% for aCGH. Combining both SNP arrays and aCGH demonstrated aneuploidy rates higher in cleavage-stage embryos (38%) compared to blastocysts (22%) ($p < 0.001$). Moreover, the cumulative pregnancy rate is reportedly better using these comprehensive chromosome screening techniques compared to FISH (62% compared to 40%) [6]. Xiong et al. [84] analyzed 169 couples (52 RobT and 117 RecT carriers): 23% of the subsequent embryos that were analyzed were unbalanced (RobT), compared to 52% for RecT. Further analysis revealed 19% of embryos from RobT carriers and 12% from RecT carriers had de novo aneuploidies for chromosomes not involved in the translocation. Idowu et al. [85] looked at 74 PGT-SR cases, establishing a statistical difference in the sex ratio of unbalanced embryos (12% male compared to 24% female, $p < 0.05$). Contrary to other studies, the percentage of unbalanced embryos did not differ between cleavage stages compared to trophectoderm biopsy groups, nor were they correlated to maternal age. Euploidy rates, however, were significantly lower in the older ($\geq$35) age group in contrast to younger counterparts (19% compared to 29%). Blastocyst embryos (42%) were more likely to be chromosomally normal than cleavage-stage (22%) embryos. Wang et al. [86] looked at 55 RobT PGT-SR cycles and 181 RecT PGT-SR cycles. Applying a regression model analysis, they found a normal/balanced rate of 42% (RobT) and 27% (RecT), similar to Idowu et al. [85] who demonstrated 37% (RobT) and 19% (RecT), respectively. In both manuscripts, cumulative pregnancy rates for embryo transfer were equivalent (44% and 43%, respectively). Zhang et al. [78] examined 11 RobT and RecT families: of 68 blastocysts, 42 were unbalanced or aneuploid with the remainder balanced or normal. Here, 13 embryos were transferred and subsequently analyzed by (amnio) prenatal diagnosis; this confirmed the initial PGT-SR. Zhang et al. [87] applied BasePhasing in two RecT families and, of 18 blastocysts, eight were unbalanced and 10 balanced/normal; two transfers followed, corroborated by amniocentesis. Beyer et al. [88] used Karyomapping in a PGT-SR setting, successfully establishing it to be applicable for distinguishing normal/balanced outcomes from unbalanced.

Taking all of the studies combined, the SNP-based approach has proven to be an effective methodology for PGT-SR, with the added bonus that it can detect balanced translocation carrier embryos. Whether such detection is morally appropriate, however, is another question and covered later.

## 14. Next-Generation Sequencing (NGS) for PGT-SR

NGS is currently the most contemporary and most widely used tool for PGT-SR. It involves the parallel genomic sequencing of a small, but representative, proportion of the whole genome (though can be targeted to a specific region). By using a DNA barcoding system to identify samples, multiple cases can be performed in the same reaction if the bioinformatic analysis is used to "bin" each sequence to the chromosomal locus and facilitate quantitative analysis. As with all DNA sequencing approaches, the cost has fallen dramatically, thereby facilitating greater uptake. NGS is relatively technically straightforward, accessible, low-cost, and has a high throughput. It detects all types of aneuploidies simultaneously, can incorporate mitochondrial DNA analysis and its greater dynamic range permits the detection of mosaicism [89,90].

Akin to aCGH and SNP analysis, this approach needs a previous whole-genome amplification step. At the time of writing, two major NGS platforms are in use for PGT-SR: semiconductor sequencing based on the detection of hydrogen ions released during DNA polymerization (Ion Torrent, Thermo-Fisher Scientific, Waltham, MA, USA), and Illumina sequencing based on sequencing by synthesis using fluorescent-labeled reversible terminators (VeriSeq, Illumina).

The most critical consideration for PGT-SR using NGS is the platform-specific resolution, in terms of the smallest segmental imbalance that can be detected. In this regard, aCGH had previously been reported as 2.5–2.8 Mb, [70,73], SNP arrays as 2.4 Mb [79], and, in the very first analysis, Ion Torrent as 5–6 Mb [91]. Cuman et al. [92] compared resolutions of aCGH compared to VeriSeq finding similar results in both (97% concordance). Nonetheless, 20% of segmental aneuploidies <20 Mb could not be detected using NGS. An initial conclusion that aCGH was the "gold standard" for PGT-SR was proposed. Yin et al. [93] compared the Illumina HighSeq2000 platform with Affymetrix SNP arrays on 38 biopsies, demonstrating a greater accuracy in the former. Additionally, Tan et al. [94] published the first appraisal of the clinical outcome of NGS PGT-SR (and PGT-A), establishing that NGS could detect some segmental aneuploidy that SNP arrays could not.

Gui et al. [95] compared diagnostic efficiencies of cleavage stage FISH with trophectoderm NGS using the Illumina HiSeq 2500 platform. The smallest fragment that could be detected was 5.1 Mb, and a 62% concordance rate was established between the two approaches. Most inconsistencies (87.5%) occurred when embryos diagnosed as unbalanced (using cleavage-stage biopsy and FISH) were found to be balanced by using trophectoderm biopsy and NGS. This study was largely the one that established that the older (cleavage-stage, FISH) approaches were no longer applicable for PGT-SR. Chow et al. [96] looked at the concordance of aCGH (Illumina 24Sure+) with NGS (Illumina VeriSeq-PGS MiSeq) in 342 embryos from 41 PGT-SR couples (38 RecT, 3 inversions); 100% concordance was established, though some segments <10 Mb could be detected using aCGH, but not NGS.

Zhang et al. [87] provided initial clinical evidence that so-called copy-number variation sequencing (CNV-Seq—Illumina HighSeq 2500 platform) was a powerful tool for PGT-SR, binning sequencing reads at smaller intervals (20 kb). Analysis of 21 PGT-SR patients (4 RobT, 17 RecT)established nearly 31% of embryos (24 day 3 and 74 day 5) were balanced for all chromosomes, with20% balanced for the translocation but aneuploid in other chromosomes. Nearly 34% of embryos were unbalanced for the translocation and 15% for both. This paper, therefore, heralded a far higher-resolution approach to PGR-SR, detecting segmental aneuploidy as low as 0.8 Mb and mosaicism as low as 20%. Moreover, Wang et al. [97] looked at 378 blastocysts from 89 RecT couples, establishing a little over 32.3% to be normal or balanced. This was, to the best of our knowledge, the largest clinical study reporting clinical pregnancy rates of around 70.5% and live birth rates of around 65.9% per embryo transfer.

In recent years, the use of NGS has expanded rapidly. Nakano et al. [98] evaluated both aCGH and NGS in PGT-SR for the purposes of avoiding recurrent miscarriage in 31 couples, 68 PGT-SR cycles, and 242 blastocysts were analyzed. Establishing a clinical pregnancy rate of 57.1% (20/35), they demonstrated the effectiveness of PGT-SR for comprehensive chromosome analysis. Similarly, Chen et al. [99] combined PGT-M, PGT-A, and PGT-SR in an NGS strategy, combining with haplotyping to produce a cost-effective universal PGT protocol, provided that a $10\times$ depth of parental and $4\times$ depth of NGS was provided.

The use of NGS has also expanded our knowledge of likely PGT-SR outcomes, empowering genetic counselors. In a study of 95 RecT and RobT 36 PGT-SR couples, Boynukalin et al. [100] looked at 532 blastocysts. Unlike most studies, they concluded the incidence of normal/balanced embryos was similar in both types (36.5% for RecT and 29.8% for RobT, $p = 0.127$). Indeed, Walters-Sen et al. [101] analyzed 238 patients, with 380 PGT-SR cycles (RecT, RobT, and inversions) to generate risk estimates for PGT-SR. Yuan et al. [102] also analyzed RecT, RobT, and inversions in 215 PGT-SR cycles (843 blastocysts). In both studies, abnormalities were higher in RecT, followed by RobT, followed by inversions, observing a clear parent-of-origin effect for RobTs and inversions. Tong et al. [103], in fact, asked whether chromosomal inversion carriers actually needed PGT-SR, evaluating rates of (an)euploidy, and mosaicism from 57 carrier couples in 71 PGT-SR cycles and 283 blastocysts. They concluded inversion type and sperm parameters in male carriers did not affect the ploidy status of embryos, with aneuploidy rates lower than predicted by modeling. They urged that male inversion carriers who had normal semen parameters and female partners <38 years old, should consider natural conception followed by a prenatal diagnosis as an alternative to PGT-SR.

Zheng et al. [104] asked whether singleton pregnancies conceived following PGT-SR were associated with a higher risk of adverse perinatal outcomes. They compared to those conceived following ICSI alone, finding no significant differences, albeit with a limited sample size and needing to be confirmed by larger studies.

When comparing different NGS platforms for the detection of segmental aneuploidy (whether de novo or arising because of an SR), greater resolution carries a higher cost. Despite this, the ability of NGS to generate massive amounts of data, plus the ever-increasing availability and cost-effectiveness of NGS, means that it is almost universally used for PGT-SR. Which specific platform to useis a matter of choice, depending on the segment size required, the sampling method used, and laboratory-specific requirements such as familiarity and brand loyalty.

## 15. Towards Universal PGT

With the recent advances in sequencing technology and bioinformatics, the number of conditions with a clearly defined genetic basis increases every day. Thus, the number of couples who want to be screened for carrier status before family planning similarly increases. This obviously brings the need for new technologies that can combine the diagnosis of more than one condition in one biopsy sample. Although it is technically possible to perform both PGT-M and PGT-A/PGT-SR in parallel, it requires careful application of different techniques such as NGS and SNP arrays sequentially with mutation and haplotype testing. However, this kind of approach is time-consuming and not suitable for widespread use due to the increase in cost. There have been attempts for developing novel methods called "Universal PGT", which can provide both rapid haplotype predictions for monogenic conditions and aneuploidy/imbalance detection in one biopsy sample. In fact, Karyomapping was designed as a universal protocol for PGT-M and PGT-A and, by extension, applicable to PGT-SR [74]. For the first time, signature patterns for a normal, balanced carrier, and unbalanced conceptuses can be seen in the Karyomapping algorithm. Other techniques were also developed such as "Haplarithmisis", "preimplantation genetic haplotyping", "Base Phasing", and more recently "Haploseek" [77,78,105]. Among them, the Haploseek method is very promising in that it combines copy-number variant (CNV) detection and whole-genome haplotype phase prediction in a cost-effective and user-friendly analysis

pipeline. This method has been validated recently for PGT-SR patients additionally which also showed that this technique is able to distinguish karyotypically normal embryos from carriers [105].

## 16. Should We Deselect or De-Prioritize Carrier Embryos?

For most cases, we can assume that "balanced" chromosome rearrangements involve neither gain nor loss of genetic material; however, this may not always be the case. Using recent technological advances such as the combination of paired-end whole-genome sequencing, it has become possible to characterize chromosomal breakpoints at a single base pair resolution [106]. Indeed, Zhai et al. [107] provided evidence that PGT-SR by NGS could discriminate normal and carrier embryos in 109 RecT and RobT carriers. Investigations in this regard suggest although the translocation may appear balanced, there may be additional or missing nucleotides because of an imprecise non-homologous end-joining process. This could result in a disease phenotype [108] in apparently balanced carrier embryos. This is something to take into consideration when prioritizing embryos for transfer. Even if we assume the rearrangement is balanced, knowing this could lead to the same reproductive problems experienced by the parents, makes the deselection of carrier embryos a consideration.

## 17. Developmental Characteristics of the Embryo and PGT-SR

The correlation between gross chromosomal errors and embryonic developmental delay or arrest is well established [109]. Whether developmental characteristics of the embryo can be used as an indicator to distinguish chromosomally unbalanced embryos in a PGT-SR setting is questionable [110]. In a very early study examining blastulation rates, no apparent selection against chromosomally unbalanced embryos was observed [110]. A much later study also found no demonstrable difference between the morphological characteristics of chromosomally normal/balanced and aneuploid embryos [111]. However, by way of contrast, Treff et al. [79] reported some demonstrable developmental differences. In this study, arrested embryos were significantly more likely to have unbalanced chromosomes in contrast to developmentally competent blastocysts. Findikli et al. [112] compared embryo morphology in translocation carriers to those of standard IVF patients, finding no differences in blastulation rates, but some differences in fertilization rates and some developmental criteria. Despite this, low sample sizes (nine RecT and six RobT) precluded definitive conclusions.

Following the advent of time-lapse technology and morphokinetic analysis, some evidence of morphokinetic differences between euploid and aneuploid embryos has emerged [113], with some parameters associated with implantation potential [113,114]. This seems to apply to numerical and segmental aneuploidies, in that the embryos carrying unbalanced translocations had delayed cleavage, delayed blastulation, and other morphokinetic parameters compared to balanced embryos [115]. Another paper compared the morphokinetic parameters of 177 chromosomally normal/balanced embryos with 250 chromosomally unbalanced ones [116]. Although significant differences were observed for some parameters, no parameter was able to predict the embryo's chromosomal status [116]. In other words, time-lapse technology may be an adjunct to PGT-SR, but should not be used as a diagnostic tool in this context. Insogna et al. [117] tested the hypotheses that, compared to PGT-A cycles (with or without PGT-M), PGT-SR cycles had a lower blastocyst conversion rate and less usable blastocysts available for transfer. They concluded that they had similar embryo development criteria but significantly fewer usable blastocysts available for transfer.

## 18. Ovarian Response and Its Relevance to PGT-SR

There are conflicting reports pertaining to the likely association between maternally derived chromosome translocations and a poor response to ovarian stimulation. Levels of estradiol on the day of human chorionic gonadotropin administration, studied in 61 cycles in 46 women with balanced chromosome translocations, were compared to a control group of 42 cycles from 32 women who had a male partner with a chromosome translocation. A higher proportion of female carriers responded poorly to ovarian stimulation compared to the control group, suggesting that ovarian response to gonadotropin stimulation may be decreased in association with chromosome translocations [118]. The study established that although not every patient carrying a chromosome translocation would be expected to have a poor ovarian response, some may be at increased risk [118]. Conversely, Dechanet et al. [119] looked at 79 cycles from 33 female chromosome translocation carriers, similarly compared with a control group of cycles with male carriers (116 cycles from 55 male carriers). No difference was observed for the following parameters: total recombinant FSH dose, number of retrieved oocytes and embryos on day three, and pregnancy rates. This study demonstrated that the response to controlled ovarian stimulation was not impaired by balanced translocation status. Whether female carriers of balanced chromosome rearrangements could be considered normal responders to controlled ovarian stimulation is not entirely certain at this stage.

## 19. Meiotic Segregation Patterns and the Number of Chromosomally Balanced Embryos Available for PGT-SR

An ability to, at least in part, predict the meiotic segregation patterns of chromosomes ahead of PGT-SR is an invaluable tool for the genetic counselor. It is well established that the probability of alternate segregation in RobTs is greater than that of RecTs (see Figures 1 and 2). Other factors may include the sex of the carrier, the position of the presence of breakpoints (particularly if near the telomere), and the type of chromosome involved [55,56,87,109,119–122].

The involvement of acrocentric (13, 14, 15, 21, and 22) chromosomes in RecT appears to impair the proportion of 2:2 (including alternate) segregations (14.6% compared to 26.0% according to Lim et al. (2008) and 39.2% compared to 60.2% according to [118]. Both sets of authors assert that the unstable nature of acrocentric chromosomes may disrupt the quadrivalent structure and thereby increase the percentage of aberrant segregation products. Wang et al. [84] reported a higher proportion of 3:1 segregation in blastocysts when acrocentric chromosomes were involved. Such phenomena may also have gender-specific differences. Studies of segregation patterns in over 2100 resultant blastocysts from 243 female and 230 male carriers (76 cases and 88 with translocations involving acrocentric chromosomes, respectively) were examined for chromosome type, carrier sex, and age. Where an acrocentric chromosome was involved, the percentage of alternate segregations (53.9% compared to 33.4%, $p < 0.0001$) was significantly greater in male carriers, with 3:1 segregation proportionally lower (6.8 compared to 16.3%, $p < 0.0001$; [119]). Indeed, the sex of the carrier influences segregational outcomes for other types of translocation, namely, 2:2 segregations, which is higher in males (60.8% compared to 52.7%, $p < 0.05$) with 3:1 and 4:0 segregations correspondingly higher in females [56]. In a study of the same year, normal/balanced embryos in RecT male carriers were higher than in female carriers (35.5 vs. 23.8%) and the percetange of 3:1 segregation patterns was greater (albeit not significantly) in female carriers [120]. Ye et al. [121] suggested that the incidence of 2:2 segregation was significantly higher in male carriers (58.2% compared to 45.0%, $p = 0.019$), a finding mirrored by Chang et al. [111] analyzing the outcomes of 66 cycles from 34 RobT carrier couples. Here, 514 blastomeres were found with a higher proportion of normal/balanced embryos in couples where the male was the carrier (32.1% compared to 27.7%). More recently [119], 154 couples with RobT (exactly equal male and female—77 each) were analyzed in 172 cycles (604 blastocysts by aCGH). Proportions of alternate, adjacent, and 3:0 segregation patterns were 68.0%, 30.6%, and 1.3%, respectively,

with alternate segregation significantly greater (82.9% compared to 55.2%; $p < 0.001$) in males. Song, H. et al. [123] examined the effect of sex and age in RecT carriers on blastocyst formation and pregnancy outcomes in 1034 PGT-SR couples. Sex-related differences were found in blastocyst formation (male > female, regardless of age) but there was no difference in fertilization rate, aneuploidy rate, clinical pregnancy rate, miscarriage rate, and live birth rate. This sex-specific selective process against unbalanced products of meiosis could be because of comparatively less strict checkpoint control in female gametogenesis [124] whereas divisional arrest results in the elimination of unbalanced meiotic segregation patterns during spermatogenesis.

Breakpoint location is another parameter possibly affecting chromosome segregation and has been studied in PGT-SR cases involving 278 embryos from 41 cycles (RecT carriers). The incidence of normal/balanced segregation patterns in RecTs with terminal breakpoints was significantly reduced (6.5% compared to 14.4%, $p = 0.005$) [121].

## 20. The Time of Biopsy and Maternal Age: No Effect on Chromosome Segregation, but Relevant to PGT-SR Nonetheless

Other factors studied include the time of the biopsy stage and, of course, maternal age. While neither appears to have a direct effect on the segregation patterns of pairing crosses per se, they nonetheless impact the % of embryos available for transfer as they do for PGT-A. The proportion of chromosomally normal embryos is associated with maternal age in PGT-SR cases, as it is in all IVF cycles [125]. This same concept also applies depending on the time of the biopsy. That is, Beyer et al. [88] clearly showed that the proportion of chromosomally normal/balanced embryos was greater at blastocyst than at the cleavage stage pointing to a selection process between days three and five in favor of karyotypically normal (or at least balanced) embryos. Among RecT carriers, the products of alternate segregation in cleavage-stage embryos (22.3%) were lower compared to blastocysts (53.1%) ($p < 0.0001$). Similarly, among RobT carriers, the proportion of product alternate segregation was significantly lower in the cleavage stage compared to blastocyst embryos (38.7% compared to 74.1; $p < 0.0001$). Xie et al. [125] also showed that the proportion of euploid (or at least balanced) embryos was significantly lower in cleavage compared to the blastocyst stage (studying both RecT and RobT). As seen in other studies, this appears to be due to selection against chromosomally abnormal embryos in the day three to five transitions.

## 21. Aneuploidy and Chromosomal Mosaicism

Structural rearrangement carriers not only suffer from segregational abnormalities, but also from abnormalities in the copy number of chromosomes or segments that are not related to the rearrangement. Aneuploidy is a frequent phenomenon in human preimplantation embryos and is the cause of a significant proportion of recurrent pregnancy loss [126]. The risk of meiotic abnormalities increases with advancing female age [127].

Aneuploidy could exist in a mosaic state where at least two chromosomally distinct cell lineages are present. Mitotic nondisjunction, anaphase lagging, trisomy rescue, formation of microscopic nuclear abnormalities (e.g., multi-nuclei and micronucleus), centriole/centrosome dysregulation, and endoreplication are the mechanisms that have been proposed in the formation of mosaicism [128,129]. Among them, mitotic nondisjunction occurs with the failure of separation of sister chromatids resulting in 3:1 segregation, i.e., a cell with monosomy and another cell with trisomy. Anaphase lagging is another frequent cause of mosaicism that occurs with the failure of a single chromatid to be incorporated into the nucleus, resulting in chromosome loss in that particular cell. In fact, the greater prevalence of monosomy over trisomy suggests anaphase lagging as the main mechanism of the formation of mosaicism in embryos [129,130]. In the context of PGT-SR, mosaic chromosome abnormalities need to be taken into account; however, whether they significantly impact live birth rates in PGT-SR cases remains to be established.

## 22. Requiem for the Interchromosomal Effect (ICE)

The possibility of an ICE was something that interested Joy Delhanty deeply. ICE can be simply described as the presence of one chromosome abnormality (such as we routinely see in PGT-SR cases) perpetuating the likelihood of further chromosome abnormalities such as (segmental or whole chromosome) aneuploidy. One of the great advantages of the practice of PGT-SR is it has allowed investigation into the basic biology of this phenomenon in human embryology. First postulated independently in 1963 by Lejeune (of Down Syndrome/Trisomy 21 fame [5]) and Lindenbaum et al. [131], who observed a higher risk of having children with Down Syndrome among carriers of rearrangements of other chromosomes. ICE can, mechanistically, be explained (at least in theory) by rearranged chromosomes impacting the segregation, pairing, and disjunction of other chromosomes in the karyotype during meiosis or subsequent mitotic divisions. Purportedly, this can result in an elevated risk of producing aneuploid gametes, perhaps due to heterosynapsis between translocated chromosomes and the sex vesicle [132]. The prospect of an ICE has been studied extensively in sperm samples of RecT and RobT carriers as well as in embryos from PGT-SR couples. Results appear to be variable and interpretations somewhat controversial. Sperm karyotyping studies have not supported the existence of an ICE in translocation and inversion carriers [23,133], nonetheless, some evidence of an ICE in carriers with compromised semen parameters has been proposed [13]. The segregation patterns of 10 chromosome pairs (chromosomes 1, 4, 9, 13, 15, 16, 20, 21, X, and Y) by sperm FISH from nine carriers were compared with three chromosomally normal men and could find no evidence of ICE in translocation carriers, who appeared fertile. For all chromosomes tested in the cohort of infertile translocation carriers, however, a correlation between poor quality sperm head and an increased aneuploidy rate was reported [13]. To the best of our knowledge, only one inversion carrier in seven cases demonstrated an ICE according to Amiel et al. [134] and an ICE has been observed in two of the seven male carriers with the same RobT [31]. In some RecT carriers, abnormal semen profiles have been reported as being more commonplace among cases that purportedly demonstrate an ICE (67%) compared with those that did not (11%) (reviewed in [23]). Therefore, some have suggested that an ICE might be may be related to infertility factors, rather than to any chromosome rearrangement directly [23]. Moreover, a possible ICE might be chromosome-specific according to Machev et al. [135], studying six male translocation carriers for chromosomes 1, 15, 16, 17, 18, X, and Y, reporting increased sperm disomy rates in three of the six chromosomes not involved in the translocation.

Godo et al. [136] highlighted the concurrent presence of aneuploidy and aberrant segregation of translocation chromosomes in the same sperm head. This study was the first to do this, suggesting that aneuploid sperm displayed significantly less 2:2 alternate segregations in the translocated chromosomes, implicating a wide general failure in meiosis I [133]. There are also contradictory results pertaining to preimplantation embryos. Gianaroli et al. [57] provided some evidence that an ICE could be the responsible mechanism that increases the proportion of embryos with abnormalities in RobT but not in RecT carriers. Other investigations found no evidence of ICE in embryos [50,137–139]. There is, however, some interesting data suggesting that RobT could trigger an ICE by inducing genetic instability in early mitotic divisions [140] when studying embryos from female carriers [141]. An analysis of 283 samples including both oocytes and cleavage-stage embryos in 44 patients using aCGH suggested a small increased risk of aneuploidy in RobT patients compared to matched non-translocation carriers. Observing this barely statistically significant result in day three embryos suggested a mitotic ICE might be present [125]. A similar effect was not seen in embryos of RecT and inversion carriers, thereby casting doubt on its overall likelihood. In a more recent study with the inclusion of a more matched control group, no ICE was observed neither in day-3 nor day-5 embryos from RobT nor RecT carriers [122] nor in heterozygote inversion carriers [140]. Perhaps the death knell has recently been dealt to the ICE by two very recent studies of PGT-SR couples. They incorporated much larger sample sizes, the latest NGS technology, matched control groups,

and robust statistical analysis. Lynch et al. [142,143] examined 1814 trophectoderm samples. No evidence was observed to support the existence of an ICE with chromosomal error rates similar in control groups. Moreover, Ogur et al. [144] provided a robust analysis of 300 couples, 443 cycles, and 1835 embryos from PGT-SR couples—the largest study to date. By including a large, matched control group making the whole study one of 5237 embryos, the cumulative de novo aneuploidy rate was actually slightly lower in PGT-SR couples compared to controls (45.6% compared to 53.4%, $p < 0.05$), but this was said to be a "negligible" association ($\varphi < 0.1$). A deeper assessment of 117,033 chromosomal pairs revealed a slightly higher individual chromosome error rate in carriers compared to controls (5.3% vs. 4.9%), which was also deemed to be a "negligible" association ($\varphi < 0.1$), despite a *p*-value of $p < 0.05$. An in-depth examination of PGT-SR patients compared to controls, therefore, suggested little or no evidence for an ICE. The results of this study (with a more appropriate control group and robust statistical analysis) not only excluded the hypothesis of ICE in their dataset but also questioned the design and interpretation of the previous work published in this context [144,145]. Indeed, in a theoretical model, Scriven [145] proposed the evidence suggests little or no confirmation for an ICE.

Scriven [145] refuted the findings of Boynukalin [100], who suggested a significant ICE effect associated with RobTs. This was given an observed higher incidence of normal/balanced embryo diagnoses having aneuploidy of an unrelated chromosome, compared to RecT. That is, a reanalysis of the same data suggests the regression coefficient of the equation (line slope) actually indicates the expected incidence of unrelated chromosomes. He also performs a similar analysis on the earlier study of Tan et al. [81], again proposing the level of aneuploidy for chromosomes other than those involved in the translocation is as expected, both for RecT and RobT.

## 23. Conclusions

Of the four forms of PGT, PGT-SR is the least well-known and perhaps the least controversial. It provides necessary and very effective treatment for many heterozygote carrier couples; avoiding repeated implantation failure, spontaneous abortions, and/or the risk of having congenitally affected children. The biggest limiting factor, however, appears to be the availability of karyotypically normal (or at least balanced embryos). The type of abnormality, the chromosomes involved, the position of breakpoints, and the sex of the carrier all seem to have an effect on the chromosome segregation pattern. While maternal age does not directly impact the segregation of the affected chromosomes, the increased number of aneuploid embryos in older women certainly impacts the management of PGT-SR couples. Of the various diagnostic techniques, NGS predominates and, although there have been prior advantages to using SNP arrays, the newer diagnostic NGS approaches will be able to do all that SNP arrays can in distinguishing carrier embryos. Detailed analysis of the carrier's embryos and literature on PGT-SR provides insights into the mechanisms of chromosome segregation in humans. This can increasingly be of direct benefit to patients as datasets get larger and can be an incredible toolkit for genetic counselors in the PGT-SR space. Since the current evidence does not support its existence, it is probably time to say "rest in peace" to the interchromosomal effect.

**Author Contributions:** D.K.G. and C.O. contributed equally to this manuscript. All authors have read and agreed to the published version of the manuscript.

**Funding:** This research received no external funding.

**Institutional Review Board Statement:** Not applicable.

**Data Availability Statement:** All data are contained within the manuscript.

**Conflicts of Interest:** The authors are employed or have their research supported by diagnostic companies (Igenomix, Cooper) that could, potentially, benefit from the publication of this manuscript.

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
