# Peer review of "PGT-SR: A Comprehensive Overview and a Requiem for the Interchromosomal Effect"

_2673-8856, doi:10.3390/dna3010004_

Round 1

Reviewer 1 Report

The authors Griffin and Ogur extensively review PGT-SR: Requiem for the Inter-Chromosomal Effect. This is a very well written review and it should be accepted for publication. It describes all types of structural rearrangements, the segregation patterns leading  to unbalanced gametes in SR carriers, the PGT strategies to avoid transfer of unbalanced embryos and the possible scenarios explaining additional chromosomal aneuploidies in embryos from SR carriers.

Only 2 minor comments

1. The authors should also mention the mechanisms leading to mosaicism

2. The authors should include the following reference in the paragraph outlining the biopsy strategies as it confirms through cell viability markers and ultrastructural assessment the safety of blastocyst biopsy and subsequent vitrification

Chatzimeletiou et al. 2022 The human embryo following biopsy on day 5 vs day 3: viability ultrastructure and spindle chromosome configurations RBMO  45(2):219-233.

1.      Chatzi

Author Response

The authors Griffin and Ogur extensively review PGT-SR: Requiem for the Inter-Chromosomal Effect. This is a very well written review and it should be accepted for publication. It describes all types of structural rearrangements, the segregation patterns leading to unbalanced gametes in SR carriers, the PGT strategies to avoid transfer of unbalanced embryos and the possible scenarios explaining additional chromosomal aneuploidies in embryos from SR carriers.

Only 2 minor comments

  1. The authors should also mention the mechanisms leading to mosaicism 

Reply: new section was added-section 21 and a few more references were added in this section too

  1. The authors should include the following reference in the paragraph outlining the biopsy strategies as it confirms through cell viability markers and ultrastructural assessment the safety of blastocyst biopsy and subsequent vitrification

Chatzimeletiou et al. 2022 The human embryo following biopsy on day 5 vs day 3: viability ultrastructure and spindle chromosome configurations RBMO  45(2):219-233. 

Reply: The information was added into section 8, the reference was added into the references list.

Reviewer 2 Report

This review describes PGT-SR, with a focus on the biology behind the structural chromosomal rearrangements, the techniques used, and the ICM. It is an interesting study, although it is a little difficult to read. The title should be changed: ICE is just a small section of the review, and it should not be mentioned in the title. I would advise to read the manuscript carefully, as many commas and full stops are missing, and many sentences are long and too informal, sometimes lacking the subject. However, the study is worthwhile publication, after minor changes.

Abstract line 16: “superseded by”

Page 1 line 43, there is a comma missing after “history”

In the introduction, please give a definition for the inter-chromosomal effect.

Line 106: please rephrase the sentence: ´Single sperm analysis by aCGH [22] is possible, as is, NGS however there are 106 significant cost implications “

Line 107: Does the word “sperm” refer to semen samples, or individual spermatozoa? Please clarify.

Line 268: misspelling in the word methods.

Line 290: there is a contraction “It’s”
Conclusion should be shortened.  

Author Response

This review describes PGT-SR, with a focus on the biology behind the structural chromosomal rearrangements, the techniques used, and the ICM. It is an interesting study, although it is a little difficult to read. The title should be changed: ICE is just a small section of the review, and it should not be mentioned in the title. 

Reply: We have revised the title giving more emphasis on the PGT-SR as an overview but kept the mention of ICE since it was one of the main motivations for the preparation of this paper and it is one of the aspects in which there is general novel insight over and above simply a review.

I would advise to read the manuscript carefully, as many commas and full stops are missing, and many sentences are long and too informal, sometimes lacking the subject. However, the study is worthwhile publication, after minor changes.

Reply: corrected 

Abstract line 16: “superseded by” 

Reply: corrected

Page 1 line 43, there is a comma missing after “history” 

Reply: corrected

In the introduction, please give a definition for the inter-chromosomal effect. 

Reply: added

Line 106: please rephrase the sentence: ´Single sperm analysis by aCGH [22] is possible, as is, NGS however there are 106 significant cost implications “ 

Reply: rephrased as requested

Line 107: Does the word “sperm” refer to semen samples, or individual spermatozoa? Please clarify. 

Reply: Revised as spermatozoa

Line 268: misspelling in the word methods.

Reply: corrected

Line 290: there is a contraction “It’s” Reply: corrected
Conclusion should be shortened.   

Reply: shortened a little bit more..

Reviewer 3 Report

This review is very comprehensive and thoroughly covers all there is to know for the common physician and IVF specialist as myself. 

Minor editing issues, the most obvious is in the last sentence of the article. I recommend  using an editing software for these minor corrections. 

Please also refer to Haplosek method, even at a glance. this is  a new developing method for PGT SR

  • DOI: 10.1038/s41436-021-01145-6

Author Response

This review is very comprehensive and thoroughly covers all there is to know for the common physician and IVF specialist as myself. 

Minor editing issues, the most obvious is in the last sentence of the article. 

Reply: deleted

I recommend using an editing software for these minor corrections. 

Reply: done

Please also refer to Haploseek method, even at a glance. this is  a new developing method for PGT SR.

  • DOI: 10.1038/s41436-021-01145-6

Reply: Thanks for this information. This method was mentioned in a new section (section 15) and the reference was added in the reference list.